# Changes in the Range of Four Advantageous Grasshopper Habitats in the Hexi Corridor under Future Climate Conditions

**DOI:** 10.3390/insects15040243

**Published:** 2024-03-30

**Authors:** Donghong Li, Huilin Gan, Xiaopeng Li, Huili Zhou, Hang Zhang, Yaomeng Liu, Rui Dong, Limin Hua, Guixin Hu

**Affiliations:** 1Key Laboratory of Grassland Ecosystem of the Ministry of Education, Engineering and Technology Research Center for Alpine Rodent Pest Control National Forestry and Grassland Administration, Pratacultural College, Gansu Agricultural University, Lanzhou 730070, China; lidonghong1204@gmail.com (D.L.); zhanghang-0214@outlook.com (H.Z.); m19896002933@163.com (Y.L.); dongrui_gsau@163.com (R.D.); hualm@gsau.edu.cn (L.H.); 2Grassland Workstation of Zhangye City, Zhangye 734000, China; ganhuilin-666@163.com; 3Grassland Technical Extension Station of Gansu Province, Lanzhou 730046, China; cvb2235@163.com (X.L.); zhhl0329@163.com (H.Z.)

**Keywords:** grasshopper, MaxEnt, climate change, suitable areas, Hexi Corridor

## Abstract

**Simple Summary:**

Grasshoppers are the most widely distributed pests in the natural grasslands of the Hexi Corridor in Gansu, northwest China. We clarified the distribution of the grasshopper suitable areas and the main environmental variables affecting the distribution of the grasshopper suitable areas, which will provide a basis for monitoring and forecasting grasshoppers in grassland. Therefore, based on the MaxEnt model, this study predicted the distribution of the four grasshoppers in their suitable areas by combining five environmental variables, namely climate, vegetation, soil, topography, and human footprint, and analyzed the main influencing factors affecting the distribution of the suitable areas. Mean annual precipitation was the main environmental variable affecting the distribution of grasshopper habitats, and the extent of the habitat of four species of grasshoppers either increased or decreased in future.

**Abstract:**

*Angaracris rhodopa* (Fischer et Walheim), *Calliptamus abbreviatus* (Ikonnikov), *Myrmeleotettix palpalis* (Zubowsky), and *Oedaleus decorus asiaticus* (Bey-Bienko) are the main grasshoppers that harm the natural grassland in the Hexi Corridor in Gansu, northwest China. In this study, the MaxEnt model was employed to identify the key environmental factors affecting the distribution of the four grasshoppers’ habitats and to assess their distribution under current and future climate conditions. The aim was to provide a basis for grasshopper monitoring, prediction, and precise control. In this study, distribution of suitable habitats for *A. rhodopa*, *C. abbreviates*, *M. palpalis*, *O. decorus asiaticus* were predicted under current and future climatic scenarios using the Maxent model. The average AUC (area under the ROC curve) and TSS (true skill statistic) values of the four grasshoppers were greater than 0.9, and the simulation results were excellent and highly reliable. The mean annual precipitation was the main factor limiting the current range of suitable areas for these four species. Under the current climate, *A. rhodopa*, *C. abbreviatus*, and *O. decorus asiaticus* were mainly distributed in the central and eastern parts of the Hexi Corridor, and *M. palpalis* was distributed throughout the Hexi Corridor, with a suitable area of 1.29 × 10^4^, 1.43 × 10^4^, 1.44 × 10^4^, and 2.12 × 10^4^ km^2^, accounting for 13.7%, 15.2%, 15.3%, and 22.5% of the total area of the grasslands in the Hexi Corridor, respectively. The highly suitable areas of *A. rhodopa*, *C. abbreviatus*, and *O. decorus asiaticus* were mainly distributed in the eastern-central part of Zhangye City, the western part of Wuwei City, and the western and southern parts of Jinchang City, with areas of 0.20 × 10^4^, 0.29 × 10^4^, and 0.35 × 10^4^ km^2^, accounting for 2.2%, 3%, and 3.7% of the grassland area, respectively. The high habitat of *M. palpalis* was mainly distributed in the southeast of Jiuquan City, the west, middle, and east of Zhangye City, the west of Wuwei City, and the west and south of Jinchang City, with an area of 0.32 × 10^4^ km^2^, accounting for 3.4% of the grassland area. In the 2030s, the range of *A. rhodopa*, *C. abbreviatus*, and *O. decorus asiaticus* was predicted to increase; the range of *M. palpalis* will decrease. The results of this study could provide a theoretical basis for the precise monitoring and control of key areas of grasshoppers in the Hexi Corridor.

## 1. Introduction

Grasslands are among the most important global ecosystems, covering about 40% of all land area and providing a wide range of ecosystem services. They not only maintain biodiversity, regulate climate, provide food, and conserve water but also serve as an essential carbon reservoir [1]. Grasshoppers are the most widely distributed and common type of insects in natural grasslands. As primary consumers in grassland ecosystems, grasshoppers participate in the material cycle and energy flow and are an important component of the food chain, occupying an important ecological position in grassland ecosystems [2].

Grasshoppers have a distribution closely correlated with the environment (climate, topography, and soil physicochemical properties) and vegetation [3]. Topography and soil primarily influence species distribution on a small scale [4]. Topography redistributes hydrothermal conditions, affecting the distribution of grasshoppers [5]. Furthermore, soil characteristics influence grasshoppers’ choice of oviposition sites, egg hatching, and mortality rates [6,7,8]. Due to variations in study areas, topographical factors, such as altitude, slope, and aspect, differ in their effects on grasshopper distribution [9,10]. Environmental factors also indirectly affect the community and spatial distribution of grasshoppers by influencing the growth and distribution of plant communities [11]. Vegetation, as a key ecological factor, not only provides food resources but also offers suitable habitats for grasshoppers [12]. On a large scale, climate-related factors are the primary determinants impacting insect distribution, and it is expected that as climate change continues, the range of suitable habitats for insects will change as well. This shift in suitable habitat ranges has important implications for the impact and management of pest species [13]. For example, warming climates effect the distribution of grasshoppers [14,15], leading to the expansion of grasshopper distribution towards higher latitudes and altitudes [16]. 

The Hexi Corridor is located deep in the interior of China’s mainland and is an important part of the Qinghai-Tibet Plateau ecological barrier in northern China [17]. As in the Inner Mongolia grasslands, *O. decorus asiaticus* (Bey-Bienko), *A. rhodopa* (Fischer et Walheim), *C. abbreviatus* (Ikonnikov), and *M. palpalis* (Zubowsky) are the dominant species of grasshopper in the Hexi Corridor [18]. *O. decorus asiaticus* (Bey-Bienko), *A. rhodopa* (Fischer et Walheim), and *M. palpalis* (Zubowsky) are more damaging to *Stipa capillata* L. and *Leymus secalinus* (Georgi) Tzvelev and *C. abbreviatus* Ikonnikov are more harmful to plants of Asteraceae *Artemisia* [19,20]. Agricultural production is mainly based on stock farming in the Hexi Corridor. Multiple species of grasshoppers occur in a year, with the peak area of grasshopper habitat reaching 1.07 million hm^2^, seriously jeopardizing the Hexi Corridor as well as the natural grasslands of the Qilian Mountains and affecting livestock development [21]. Therefore, determining the habitat suitability of the four grasshoppers on the Hexi Corridor is critical to developing methods to reduce the impact of future outbreaks.

Using grasshopper distribution data and environmental variables to model the ecological niche of grasshoppers and assess habitat quality in grasshopper suitable areas can help develop scientifically sound pest control measures [22,23]. Currently, widely used species distribution models (SDMs) include random forest (RF) [24], logistic regression model [25], generalized linear model (GLM) [26], ecological niche factor analysis (ENFA) [27], Bioclimate Analysis and Prediction System (BIOCLM) [28], and maximum entropy (MaxEnt) [29,30]. Among them, due to its advantages of being unaffected by sample size, its simple operation, and its high predictive accuracy [31], MaxEnt is widely applied in different research areas, such as the conservation of animal and plant habitats [32,33], the protection of endangered species [34], the assessment of biological invasion [35,36], and in disease prevention and surveillance [37,38,39]. Many scholars have researched the suitable habitats and influencing factors of grasshoppers in China using MaxEnt. Due to variations in the study area, time, and selected environmental variables, the main environmental factors affecting grasshopper distribution differ. However, the predicted results of the models were accurate [40,41,42,43].

Currently, there are limited studies on the distribution of suitable areas for grasshoppers in the Hexi Corridor as well as in the Qilian Mountains. Lv et al. [44] made predictions about suitable habitats for grasshoppers in the alpine grasslands of the Qilian Mountains. On a small scale, numerous scholars have conducted research on the species composition [2] and quantitative characteristics [18] of grasshoppers in the Qilian Mountains, as well as their relationships with vegetation communities [45], topography [46], temperature, and rainfall [47]. Due to the influence of climatic and topographic conditions, the grassland ecosystem of the Qilian Mountains is complex and diverse, providing varied and suitable habitat conditions for various grasshopper species, making grasshopper control difficult [48]. Traditional grasshopper monitoring mainly collects data manually, which is time consuming and labor intensive, and grasshoppers are widely distributed, especially in remote areas of the steppe, which makes field investigation difficult [41].

Using the MaxEnt model to combine grasshopper distribution data and environmental data, this study aims to: (1) determine the suitable areas of *A. rhodopa*, *C. abbreviatus*, *M. palpalis*, and *O. decorus asiaticus* in the natural grasslands of the Hexi Corridor; (2) identify key environmental factors affecting the distribution of the four grasshoppers in the suitable areas; and (3) analyze changes to the suitable areas of the four grasshoppers under future climate conditions. This study will provide theoretical guidance for the monitoring and prediction of grasshoppers in the Hexi Corridor and the northern Qilian Mountains.

## 2. Materials and Methods

### 2.1. Study Area

This study was conducted in the Hexi Corridor (Figure 1), which is in the northern foothills of the Tibetan Plateau and the southern edge of the Mongolian Plateau. It is a transitional zone where the two major plateaus of Mongolia and Qinghai are intertwined. The climate is affected by the geographic latitude, changes in the altitude gradient on the Tibetan Plateau, and the continental climate on the Mongolian Plateau, which has diversified and drastically changed [49]. The climate is temperate continental, with an annual precipitation amount of 40–300 mm and an annual temperature of 6.2–9.0 °C. The annual evaporation in most areas exceeds 1500 mm. The altitude is 1300–4200 m. The forests and grasslands of the Qilian Mountains collectively form an ecological barrier in the northwest region of China [50,51]. Due to the strong folded uplift of the Qilian Mountains and the substantial subsidence of the corridor zone, the area has evident vertical zonation, resulting in various vegetation types. Among them, natural grasslands are the predominant vegetation type, covering approximately 53% of the total area [52,53]. From low to high altitude, the grassland types include low plain meadow, swamp meadow, temperate desert, temperate grassland desert, temperate desert steppe, temperate steppe, mountain meadow, alpine meadow, alpine scrub meadow, alpine steppe, and alpine desert. The soil types include montane gray calcium soil, montane chestnut calcium soil, montane grey brown soil, scrub meadow soil, and chilly desert soil [51,54,55].

### 2.2. Data Acquisition and Processing

#### 2.2.1. Grasshopper Survey Data

Grasshopper were investigated according to the agricultural industry standard of the People’s Republic of China (NY/T1578-2007 Grasshopper investigation specification). From June to August 2021–2023, a regional survey method was used, with each elevation of 200 m divided into a gradient along the direction of low elevation to high elevation. One elevation gradient could be used as a sample point, and the distance between sample points was not less than 100 m. The survey route covered all major geomorphological units and grassland types. The presence and location of grasshoppers were recorded during the survey but no data on grasshopper disappearances were recorded. Field surveys were conducted using GPS to record latitude, longitude, and elevation, and four grasshoppers (*A*. *rhodopa*, *C*. *abbreviatus*, *M*. *palpalis*, and *O*. *decorus asiaticus*) were selected from the natural steppes of the Hexi Corridor for the study.

We collected 105 data on the occurrence of *A. rhodopa*, 91 on the occurrence of *C. abbreviatus*, 181 on the occurrence of *M. palpalis*, and 102 on the occurrence of *O. decorus asiaticus*. To avoid over-fitting due to sampling deviation, and minimize the effects of spatial autocorrelation [56], screening was conducted with the “Create Fishnet” tool in ArcGIS 10.2, and a grid of 5 km × 5 km was established to ensure that there was only one distribution point within each grid. Ultimately, there were 68 distribution points for *A*. *rhodopa*, 58 distribution points for *C*. *abbreviatus*, 92 distribution points for *M*. *palpalis*, and 61 distribution points for *O*. *decorus asiaticus*, with a total of 279 distribution points. The geographic information for the four grasshoppers was saved in CSV format for MaxEnt modeling.

#### 2.2.2. Environmental Variables

Climate data were obtained from the World Climate Database (World Clim version 2.0, http://www.worldclim.org/, accessed on 5 October 2022) for 19 biological variables with a spatial resolution of 1 km (Table 1). The BCC-CSM2-MR climate model under the 6th International Coupled Model Intercomparison Program (CMIP6) climate model was selected for the future climate variables, using data from 2021 to 2040 under four shared socio-economic pathways (SSPs) scenarios: SSP126 (low GHG emissions: carbon dioxide emissions fall to net zero around 2075); SSP245 (intermediate GHG emissions: carbon dioxide emissions will remain at current levels until 2050 and then decline, but will not reach net zero by 2100); SSP370 (high GHG emissions: carbon dioxide emissions will double by 2100); and SSP585 (very high GHG emissions: carbon dioxide emissions will triple by 2075). BCC-CSM2-MR significantly improved the simulation of the climate distribution of mean annual precipitation in China compared to CMIP5 from the previous generation [57].

The land surface temperature and normalized difference vegetation index (NDVI) were obtained from MODIS (https://modis.gsfc.nasa.gov/, accessed on 27 November 2023), using average values from 2013 to 2022 with a spatial resolution of 1 km. The grassland type from Gansu Provincial Grassland Technology Extension General Station was converted to raster using the ArcGIS 10.2 conversion tool, with spatial resolution resampled to 1 km. The topographic variables were obtained from the elevation downloaded from the Geospatial Data Cloud (https://www.gscloud.cn/, accessed on 31 August 2023) with a resolution of 1 km, and elevation, slope, and slope direction were calculated using ArcGis 10.2. Soil was obtained from the Harmonized World Soil Database (HWSD, https://gaez.fao.org/pages/hwsd, accessed on 19 November 2023), made publicly available by the Food and Agriculture Organization of the United Nations (FOA), at a resolution of 1 km. Human footprint (HFP) was obtained from the Fig Share repository (https://figshare.com/, accessed on 20 November 2023),) at a resolution of 1 km. We used an annual human footprint dataset for the global landmass from 2000 to 2018, published online in Scientific Data by Mu et al. [58]. The human footprint was mapped using eight variables reflecting human pressures (built environment, population density, nighttime lighting, cropland, rangeland, roads, railroads, and navigable waterways). It was found that the inclusion of human activity intensity data for modeling and prediction on a large-scale prediction of the area and range of a species can improve the accuracy of prediction results [59].

#### 2.2.3. Environment Variable De-Correlation

Environmental variables are highly spatially correlated; this may lead to the overfitting of the model and ultimately affect the prediction results [34]. Therefore, 19 biological variables of grasshopper distribution points were extracted using multi-value extraction to points in ArcGIS 10.2. The correlations among the 19 biological variables were tested using Pearson’s correlation analysis in SPSS 24, and variables with absolute values of correlation coefficients |r| < 0.8 for environmental variables were retained [60].

### 2.3. MaxEnt Model Runs

Using ArcGIS10.2 to remove the spatial correlation after grasshopper coordinate points, the environmental variables with high correlation were converted into ASCII format and then imported into MaxEnt3.4.4. Of the distribution point data, 75% were used for modeling and 25% were used for validation. There were 10,000 iterations. The run was repeated 30 times, and the average value was selected as a prediction of the distribution of grasshoppers. The results were exported as logistic models and saved in ASC format [61]. The receiver operating characteristic curve (ROC) test simulation prediction results, ROC curve, and horizontal coordinate axis were used to determine the area under the ROC curve (AUC). The accuracy of the AUC value was between 0 and 1. The higher the AUC value, the higher the accuracy of the model. The prediction results were classified as failure (0.5–0.6), poor (0.6–0.7), general (0.7–0.8), good (0.8–0.9), and excellent (0.9–1.0) [62]. The true skill statistic (TSS) value was in the range of −1 to 1. The higher the TSS value, the greater the consistency of the observed values with the predicted values, and the better the model. The greater the model effect, the lower the TSS value, the worse the consistency, and the worse the model prediction effect [63]. The potential distribution of grasshoppers obtained from the MaxEnt model was transformed into raster form using ArcGIS10.2, and the simulation results were reclassified (Reclassify) using the natural intermittent point classification method (Jenks). The grasshoppers’ fitness zone was classified into non-fitness zone, low fitness zone, medium fitness zone, and high fitness zone, and the fitness zone rank of grasshoppers was obtained using the Arc GIS10.2 distribution map. At the same time, the spatial statistics function of ArcGIS10.2 was utilized to calculate the areas of different suitable zones.

## 3. Results

### 3.1. Accuracy of the MaxEnt Model

The prediction results of the MaxEnt model showed that the mean AUC values of *A. rhodopa*, *C. abbreviatus*, *M. palpalis*, and *O. decorus asiaticus* were 0.958, 0.949, 0.929, and 0.946, respectively, and that the mean TSS values were 0.972, 0.970, 0.954, and 0.971, respectively. The mean AUC and TSS values of the four grasshoppers were greater than 0.9, indicating that the model prediction results had high reliability and could reasonably simulate the distribution of the four grasshopper species (Table 2).

### 3.2. Effects of Major Environmental Variables on the Distribution of Grasshoppers

Among the five types of variables, namely climate, vegetation, topography, soil, and human footprint, climate made the greatest cumulative contribution to *A. rhodopa*, *C. abbreviatus*, *M. palpalis*, and *O. decorus asiaticus*, accounting for 53.8%, 46.9%, 42.2%, and 47.6%, respectively. Vegetation variables accounted for 30.8%, 10.2%, 34.7%, and 18.8%, respectively, and topography variables accounted for 4.0%, 6.8%, 8.3%, and 5.7%, respectively. Soil variables accounted for 6.0%, 16.2%, 2.4%, and 19.0%, respectively, and human footprint variables accounted for 5.8%, 19.9%, 12.4%, and 8.9%, respectively. Among the climatic variables, the average annual rainfall (Bio12) contributed the most to *A. rhodopa*, *C. abbreviatus*, and *O. decorus asiaticus* (35.5%, 33.5%, and 33.6%, respectively). The NDVI contributed the most to *M. palpalis*, with 31.8%, while the average annual rainfall (Bio12) contributed to *M. palpalis* with 9.8%(Table 3).

### 3.3. Distribution and Size of Suitable Areas for Grasshoppers in the Current Climate

The grassland area of the Hexi Corridor is about 9.4 × 10^4^ km^2^. The suitable areas of *A. rhodopa*, *C. abbreviates*, and *O. decorus asiaticus* were mainly located in the middle and eastern parts of the Hexi Corridor, with total suitable areas of 1.29 × 10^4^, 1.43 × 10^4^, and 1.44 × 10^4^ km^2^, respectively. The suitable area of *M. palpalis* was located throughout the Hexi Corridor, with a total suitable area of 2.12 × 10^4^ km^2^ (Figure 2). Under the current climatic background, suitable areas for *A. rhodopa*, *C. abbreviatus*, *M. palpalis*, and *O. decorus asiaticus* accounted for 13.7%, 15.2%, 22.5%, and 15.3% of the grassland area, respectively (Figure 3). The highly suitable areas for *A. rhodopa*, *C. abbreviates*, and *O. decorus asiaticus* were mainly distributed in the central and eastern parts of Zhangye City, the western part of Wuwei City, and the western and southern parts of Jinchang City, with areas of 0.20 × 10^4^, 0.29 × 10^4^, and 0.35 × 10^4^ km^2^, respectively, accounting for 2.2%, 3%, and 3.7% of the grassland area, respectively. The highly suitable area for *M. palpalis* was mainly located in the southeastern part of Jiuquan City; Zhangye City; the western, central, and eastern parts of Wuwei City; and the western and southern part of Jinchang City; with an area of 0.32 × 10^4^ km^2^, accounting for 3.4% of the grassland area.

### 3.4. Potential Distribution of Grasshoppers under Future Climates

The extent of the main habitats of the four grasshoppers did not change under future climate conditions. Over time, the fitness zones of *O. decorus asiaticus*, *C. abbreviates*, and *A. rhodopa* all increased, and the range of *M. palpalis*’ fitness zone decreased. Under future climate conditions, the total area of the suitable zone of *O. decorus asiaticus* under SSP126, SSP245, SSP370, and SSP585 accounted for 15.2%, 14.7%, 15.6%, and 15.6% of the grassland area, respectively (Figure 4). Compared to the area of the current suitable zone, the area of *O. decorus asiaticus* under SSP126 decreased, and the area of *O. decorus asiaticus* under SSP245, SSP370, and SSP585 increased. *Calliptamus abbreviates* increased its suitable area under SSP126, SSP245, SSP370, and SSP585, which accounted for 15.5%, 15.4%, 15.7%, and 15.6% of the grassland area, respectively, with the largest increase of 1.48 × 10^4^ km^2^ under SSP370. The fitness zone of *A. rhodopa* decreased in size under SSP126, increased in size under both SSP245 and SSP370, and did not change in size under SSP585. *Myrmeleotettix palpalis* had the largest suitable area, which was reduced under SSP126, SSP245, SSP370, and SSP585, accounting for 22.2%, 22.0%, 22.1%, and 22.3% of the grassland area, respectively, with the largest reduction under SSP245, which was reduced by 0.5 × 10^4^ km^2^ (Figure 2).

## 4. Discussion

### 4.1. Selection of Distribution Points and Variables for the MaxEnt Model

In this study, the average AUC and average TSS of *A. rhodopa*, *C. abbreviatus*, *M. palpalis*, and *O. decorus asiaticus* were greater than 0.9, indicating that the model had high accuracy for the simulation of the fitness zones of the four grasshopper species. Since grasshopper population size and distribution have been in a state of dynamic change throughout their life history, and because the MaxEnt Model assumes that the species are in equilibrium, the accuracy of the model may be reduced [41]. The Hexi Corridor is an important ecological security barrier in China, as well as a strategic corridor for transportation, energy, telecommunication, and logistics, and an important section of the “One Belt, One Road” construction. Due to ongoing economic development and frequent human activities, the natural grassland in the area has been destroyed. Vegetation cover, which provides a suitable habitat for grasshoppers, has particularly declined [64,65,66]. Therefore, the human footprint was selected to explain the effects of human activities on grasshoppers, and increasing human activity variables improved the model predictions [59]. In this study, adult grasshoppers were used as the object, and the effects of environmental factors on grasshopper suitable areas during the incubation period were not considered; therefore, further research is needed to improve the accuracy of model predictions.

### 4.2. Influence of Environmental Variables on Grasshopper Distribution

The MaxEnt model showed that climate was the main factor affecting the distribution of *A. rhodopa*, *C. abbreviatus*, *M. palpalis*, and *O. decorus asiaticus*. Among climatic factors, the mean annual precipitation (Bio12) contributed the most to the distribution of *A. rhodopa*, *C. abbreviatus*, *M. palpalis*, and *O. decorus asiaticus*, while the mean annual temperature (Bio1) had a lower contribution. This was different from the results of studies in other regions. For instance, on the Mongolian Plateau, the key environmental variables affecting the distribution of grasshoppers was vegetation [41], probably due to different climatic conditions. The Hexi Corridor is in the interior of Asia and has a mainly arid or semi-arid climate with a dry climate, sparse precipitation, and high temperatures, while the Mongolian Plateau is in Central Asia and has a cold climate with relatively high precipitation and low temperatures. In this study, the low contribution of annual mean temperature and land surface temperature to grasshoppers was notable. It has been shown that grasshopper infestations are highly correlated with precipitation and have a very weak correlation with temperature [67]. In addition, grasshoppers are prone to forming outbreaks under extreme climatic conditions, especially during dry and warm years [68]. Stunting occurs in the eggs of certain grasshoppers in response to adverse environmental conditions that are inadequate for the survival of more grasshoppers [69]. During the investigation period of this study, the Hexi Corridor was dry for three consecutive years, and grasshopper colonies were more abundant only on a small regional scale. Therefore, it is hypothesized that a major grasshopper outbreak may occur after the drought ends.

Changes in the growth of grassland vegetation caused by seasonal and climatic conditions can alter the grasshopper community structure [70]. The occurrence of grasshoppers is synchronized with changes in vegetation growth [71]. In this study, vegetation variables were second only to climate variables for *A. rhodopa*, *C. abbreviatus*, *M. palpalis*, and *O. decorus asiaticus*. Among them, NDVI was the main vegetation factor influencing the distribution of grasshoppers. Vegetation is an important factor affecting the distribution of grasshoppers, and grasshoppers need to feed on vegetation to obtain energy and complete their life cycle, so vegetation directly determines the growth, development, and reproduction of grasshoppers [72]. Among the four species of grasshoppers studied, vegetation has the least influence on the distribution of *C. abbreviatus*, probably because *C. abbreviates*, which have a wider dietary pattern, not only feeds on grasses but also on plants of *Artemisia*, Asteraceae [20]. As a result, *C. abbreviatus* could have outbreaks due to sufficient food and require focused monitoring.

Topographic and soil factors are important in the study of small-scale patterns of species [73,74]. Among them, topography is a multidimensional variable that includes factors such as elevation, slope direction, slope gradient, slope shape, and slope position, which not only determine the spatial distribution of light, heat, water, and soil but also directly affect the distribution of plant and animal communities and the formation of population patterns [75]. The northern Qilian Mountains are steep and have an overall slope. In this study, the four grasshoppers were mainly distributed at a slope of about 71°, and the slope contributed 2.8%, 4.8%, 5.9%, and 3.8% to the distribution of *A. rhodopa*, *C. abbreviatus*, *M. palpalis*, and *O. decorus asiaticus*, respectively, which was an important topographic factor influencing the distribution of the four grasshoppers in suitable areas. Elevation contributed less to *C. abbreviatus* and *O. decorus asiaticus* but was more important than slope and direction. It also contributed less to *A. rhodopa* and *M. palpalis*, both in terms of contribution and importance. At an altitude of about 2700 m, the distribution probability of all four grasshoppers was higher. The environment is wetter and colder above an altitude of 3000 m. Below an altitude of 1500 m it is windy and sandy, mostly desert, with drastic changes in heat and cold, dryness, and little rain, all of which are unfavorable conditions for grasshopper survival. Soil texture, water content, salinity, and pH affect grasshoppers’ choice of spawning sites, egg development, and hatching [76]. In addition, soil indirectly affect grasshopper distribution by influencing vegetation type and growth [77]. In this study, soil moisture contributed 5.4%, 15.9%, 1.4%, and 18.8% to the distribution of *A. rhodopa*, *C. abbreviatus*, *M. palpalis*, and *O. decorus asiaticus*, respectively, and was the main soil variable associated with grasshopper distribution.

The global human footprint mapped by eight variables—built environment, population density, nighttime lighting, cropland, pasture, roads, railroads, and navigable waterways—were used in this study to reflect the impact of human activities on grasshopper distribution. Livestock grazing is one of the most important human activities affecting grasshopper distribution, and grazing alters the grasshopper community structure [78,79]. Hao et al. [80] found that grazing reduced the diversity of grasshopper species in arid regions but significantly increased the number of dominant species. Cease et al. [81] also found that overgrazing promoted outbreaks of grasshoppers. In this study, the human footprint contributed 5.8%, 19.9%, 12.4%, and 8.9% to the distribution probabilities of *A. rhodopa*, *C. abbreviatus*, *M. palpalis*, and *O. decorus asiaticus*, respectively, and the distribution probabilities of the four grasshoppers increased significantly as the intensity of the human footprint increased. Therefore, human activities increase the probability of grasshopper occurrence. It has been found that nighttime lights attract grasshoppers [82]. With the development of the economic zone in the Hexi Corridor, the nighttime lighting brought about by infrastructure construction has also become a factor affecting the distribution of grasshoppers. Nighttime lights may prolong the photoperiod and affect grasshopper spawning. Hiroshi [83] found that grasshopper females laid non-dormant and dormant eggs under long and short photoperiods. Thus, human activities have a significant impact on grasshopper growth, development, reproduction, and habitat selection.

The importance of individual environmental factors varies for different grasshoppers, and there is a hierarchy between environmental variables. In addition, multiple species of grasshoppers could occur simultaneously in grasslands. When predicting grasshopper occurrence, the dominant species of a grasshopper swarm should be the primary monitor.

### 4.3. Changes in the Distribution of Grasshopper Habitat Areas under Future Climate Scenarios

The MaxEnt model was used to predict the distribution ranges of the fitness zones of *O. decorus asiaticus*, *C. abbreviatus*, *A. rhodopa*, and *M. palpalis* under four emission scenarios (SSP126, SSP245, SSP370, and SSP585) in the 2030s. The results showed that under the four discharge scenarios, the size of the suitable area for *C. abbreviatus* expanded relative to the current situation; the size of the suitable area for *A. rhodopa* was less under SSP126, and unchanged under SSP585; the size of suitable area for *O. decorus asiaticus* decreased under SSP126 and SSP245; and *M. palpalis*’ suitable area decreased in all scenarios. Meynard et al. [84] predicted that the distribution area of *Schistocerca gregaria* would change under extreme climate change scenarios, with a decrease in the range contraction of the northern subspecies *Schistocerca gregaria gregaria* and an increase in the range contraction of the southern subspecies *Schistocerca gregaria flaviventri*. Among all variables, climatic variables, especially precipitation, were closely related to the distribution of the four grasshoppers. The MaxEnt model predicted the geographic distribution of dominant grasshoppers in the Hexi Corridor under the future climate conditions, which can help monitor grasshopper occurrence and provide a reference for prevention and control in key areas.

### 4.4. Monitoring and Protection

Based on the findings of this study, monitoring should be focused on the current medium and high suitable areas of grasshoppers, including density, composition, and habitat environment. In this study, climate, vegetation, soil, topography, and human footprints all have influence on the distribution of grasshopper suitable areas. By monitoring environmental variables and the densities and distribution of grasshopper in suitable areas, relationships between environmental variables and grasshopper occurrence can be established to effectively predict and accurately control them. As the climate warms, the ranges of most species will expand [85,86,87], while the ranges of some species with special habitat needs will be restricted [88]. This is mainly due to the migration of species to higher latitudes and altitudes as the temperature increases. Appropriate habitat protection and management are essential for the survival of grasshopper populations, especially changes in habitat structure caused by human or natural factors, and to minimize the impact of land-use changes to ensure the survival of the species.

## 5. Conclusions

In this study, we used four grasshopper species to predict the distribution of the dominant grasshopper niche in the grasslands of the Hexi Corridor under current and future climates based on the MaxEnt model. The main environmental variable affecting the distribution of the grasshopper niche was rainfall, followed by vegetation, soil, and topography. The range of the grasshoppers’ niche will be altered under future climate conditions. Based on the results of this study, further monitoring of grasshoppers in Zhangye City is needed.

## Figures and Tables

**Figure 1 insects-15-00243-f001:**
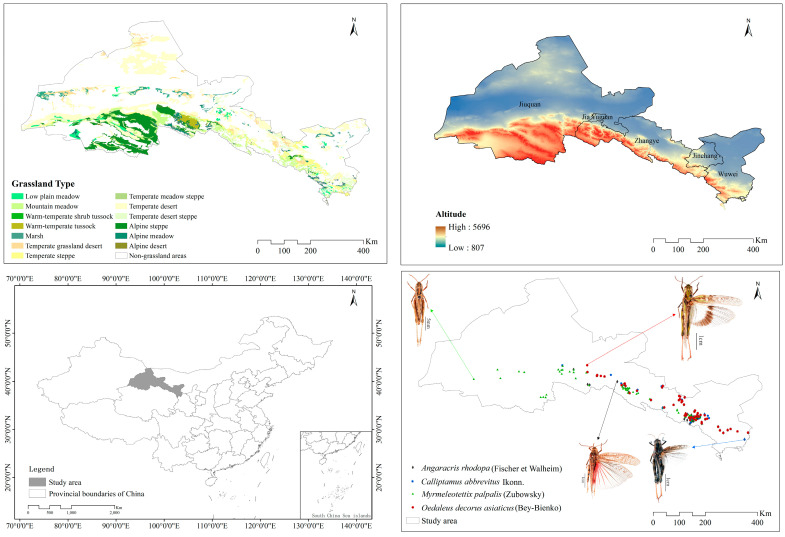
Distribution points of the grasshoppers.

**Figure 2 insects-15-00243-f002:**
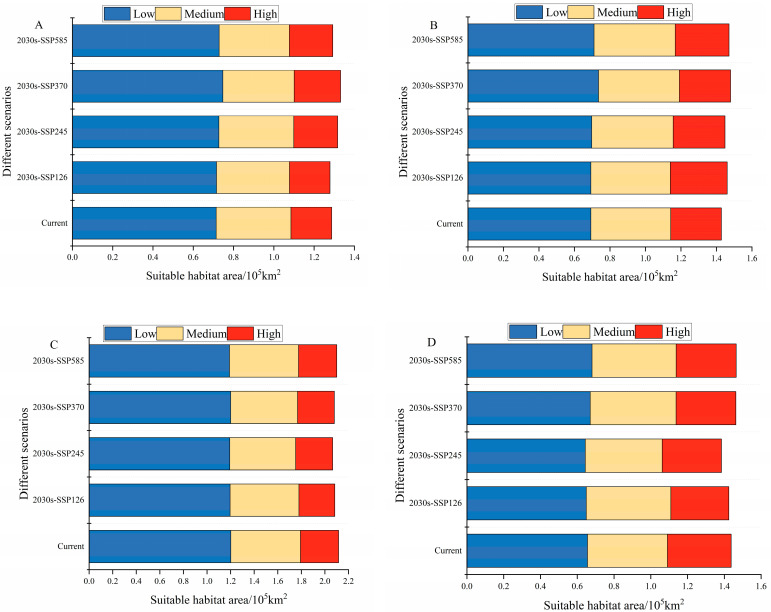
Changes in the size of suitable areas for (**A**) *Angaricus rhodopa*, (**B**) *Calliptamus abbreviates*, (**C**) *Myrmeleotettix palpalis*, and (**D**) *Oedaleus decorus asiaticus* under current and future climate scenarios.

**Figure 3 insects-15-00243-f003:**
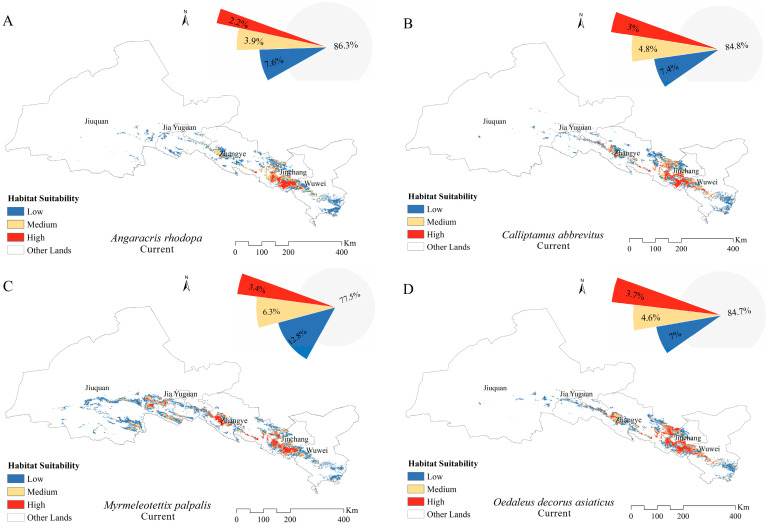
Distribution of suitable areas and proportion of suitable habitats for (**A**) *Angaricus rhodopa*, (**B**) *Calliptamus abbreviates*, (**C**) *Myrmeleotettix palpalis*, and (**D**) *Oedaleus decorus asiaticus* under the current climate scenario.

**Figure 4 insects-15-00243-f004:**
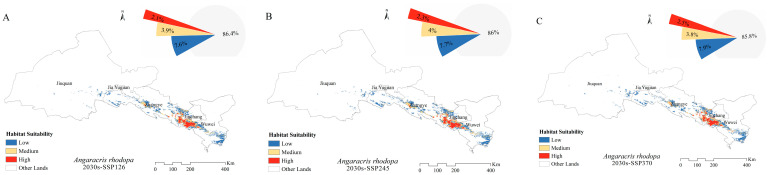
Distribution of suitable areas and proportion of suitable habitats for (**A**–**D**) *Angaricus rhodopa*, (**E**–**H**) *Calliptamus abbreviates*, (**I**–**L**) *Myrmeleotettix palpalis*, and (**M**–**P**) *Oedaleus decorus asiaticus* in the 2030s.

**Table 1 insects-15-00243-t001:** Environment variables.

Type	Code	Variable Name
Climatic	Bio1	Annual mean temperature
	Bio2	Mean diurnal range (monthly mean (max temp minus min temp))
	Bio3	Isother mality (BIO2/BIO7) (×100)
	Bio4	Temperature seasonality (standard deviation × 100)
	Bio5	Max temperature of warmest month
	Bio6	Min temperature of coldest month
	Bio7	Temperature annual range (BIO5 minus BIO6)
	Bio8	Mean temperature of wettest quarter
	Bio9	Mean temperature of driest quarter
	Bio10	Mean temperature of warmest quarter
	Bio11	Mean temperature of coldest quarter
	Bio12	Annual precipitation
	Bio13	Precipitation of wettest month
	Bio14	Precipitation of driest month
	Bio15	Precipitation seasonality (coefficient of variation)
	Bio16	Precipitation of wettest quarter
	Bio17	Precipitation of driest quarter
	Bio18	Precipitation of warmest quarter
	Bio19	Precipitation of coldest quarter
	LST	Land surface temperature
Vegetation	NDVI	Normalized difference vegetation index
	GT	Grassland type
Topographical	Elevation	Elevation
	Slop	Slop
	Aspect	Aspect
Soil	AWC	Soil available water content
	ST	Soil type
	PH	T_PH
Human activity	HFP	Human Footprint Index

**Table 2 insects-15-00243-t002:** Average AUC and TSS values of the model run.

Time	Emission Scenarios	*A. rhodopa*		*C. abbreviatus*		*M. palpalis*		*O. decorus asiaticus*	
		Training AUC	Test AUC	Training AUC	Test AUC	Training AUC	Test AUC	Training AUC	Test AUC
Current		0.972	0.958	0.970	0.949	0.954	0.929	0.971	0.946
2021–2040	SSP126	0.972	0.953	0.971	0.944	0.955	0.932	0.973	0.947
	SSP245	0.973	0.955	0.967	0.942	0.956	0.933	0.974	0.950
	SSP370	0.972	0.952	0.972	0.943	0.956	0.931	0.974	0.951
	SSP585	0.972	0.954	0.970	0.946	0.956	0.929	0.972	0.946

**Table 3 insects-15-00243-t003:** Relative contributions of variables to grasshoppers.

*A. rhodopa*			*C. abbreviatus*			*M. palpalis*			*O. decorus asiaticus*		
Variables	Percent Contribution (%)	Cumulative Contribution Rate (%)	Variables	Percent Contribution (%)	Cumulative Contribution Rate (%)	Variables	Percent Contribution (%)	Cumulative Contribution Rate (%)	Variables	Percent Contribution (%)	Cumulative Contribution Rate (%)
Bio12	35.5	35.5	Bio12	33.5	33.5	NDVI	31.8	31.8	Bio12	33.6	33.6
NDVI	30.3	65.8	HFP	19.9	53.4	HFP	12.4	44.2	AWC	18.8	52.4
Bio7	8.5	74.3	AWC	15.9	69.3	Bio12	9.8	54	NDVI	18.6	71
HFP	5.8	80.1	NDVI	9.7	79	Bio1	9.5	63.5	Bio7	9.3	80.3
AWC	5.4	85.5	Bio7	7.3	86.3	Bio2	8.2	71.7	HFP	8.9	89.2
Bio15	4.1	89.6	Slop	4.8	91.1	Bio7	8.1	79.8	Slop	3.8	93
Slop	2.8	92.4	Bio3	2.7	93.8	Slop	5.9	85.7	Bio15	2.8	95.8
LST	2.5	94.9	Bio15	2	95.8	Bio19	5.6	91.3	Elevation	1.5	97.3
Bio2	1.9	96.8	Elevation	1.3	97.1	GT	2.9	94.2	Bio2	0.9	98.2
Aspect	1.1	97.9	Bio1	0.9	98	Aspect	1.8	96	Bio1	0.8	99
Bio1	0.7	98.6	Aspect	0.7	98.7	AWC	1.4	97.4	Aspect	0.4	99.4
GT	0.5	99.1	GT	0.5	99.2	ST	0.9	98.3	GT	0.2	99.6
Bio19	0.4	99.5	Bio2	0.3	99.5	LST	0.6	98.9	Bio19	0.2	99.8
Bio4	0.2	99.7	ST	0.2	99.7	Elevation	0.6	99.5	ST	0.1	99.9
ST	0.1	99.8	PH	0.1	99.8	Bio14	0.3	99.8	PH	0.1	100
Elevation	0.1	99.9	LST	0.1	99.9	PH	0.1	99.9	LST	0	100
PH	0.1	100	Bio4	0.1	100	Bio4	0.1	100			

## Data Availability

The data are not publicly available because the data need to be used in future work.

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
