# Peer review of "Changes in the Range of Four Advantageous Grasshopper Habitats in the Hexi Corridor under Future Climate Conditions"

_insects, 2024, doi:10.3390/insects15040243_

Round 1
Reviewer 1 Report
Comments and Suggestions for Authors
Please see attached document with line-by-line comments

Comments on the Quality of English LanguageIn general, the manuscript is OK. Several sentences and paragraph can be edited for clarity
Reviewer 2 Report
Comments and Suggestions for Authors
Dear Authors,
The manuscript appears to be technically sound. In the paper, the focus was on background research in the field of modelling; however, there was a lack of clear information regarding why these insect species were addressed. In reality, the reader only receives information that the insect is a pest, but the manuscript does not provide details about the extent and scale of these damages. It is not even known which plants are mainly affected. It is worthwhile to supplement this information to ensure that the paper is comprehensive and does not require the reader to speculate about the reasons for conducting the research.
I cannot find any information source showing that the chosen species are dominant in the natural grassland of the Hexi Corridor and the northern Qilian Mountains. I cannot find any info about preliminary studies or cited papers suggesting it.
In hole manuscript even not once is giving full Latin name of research object appeared. For example, “Calliptamus abbreviatus Ikonnikov, 1913” should be given, not only “Calliptamus abbreviatus”. Check it and add full names for each species in the manuscript.
Lines 101, 266, 284, 361. There is no such orthopteran like Angaricus rhodopa, correct name is Angaracris rhodopa (Fischer de Waldheim, 1836).
Line 127. I tried to find the specification NY/T1578-2007 but it is unevaluable to download in English. I suggest to describe it in one paragraph or place working link to pdf in English also in literature.
Line 239. Lacking number “3” before the name of subsection.
Figure 4. Maps are very small. I know that in the electronic version (pdf), it is possible to zoom in; however, after printing, descriptions are not visible. I suggest placing bigger maps, a maximum of two maps, side by side in the text or choosing some of them, the most important, and another place as appendices.
After reading the paper and analysing the errors made, it seems that the author team lacks an entomologist who would outline the research background and indicate the reasons for conducting the research. In its current form, the descriptions focus on spatial modelling and technical matters, but there is a lack of discussion on the environmental and economic consequences and information about the insects themselves.
I suggest adding an entomologist familiar with the subject matter of these insects to the team. This entomologist could provide descriptions of their biology and significance, allowing for the expansion of the discussion towards their economic importance, not just their spatial relevance.
The paper shows potential, however, in order for it to be fully valuable and maintain a high level appropriate for journal standards, it is necessary to address the identified shortcomings.
Best regards
Reviewer 3 Report
Comments and Suggestions for Authors
This research conducted species distribution modelling on Grasshoppers to understand where they might impact grasslands in China under future predicted climate change. The study concentrated on this species given grasslands are an important habitat type in this region.
Introduction and Methods
The introduction is thorough in providing background to climate modelling and the model chosen. However, much if not the entire first paragraph should be placed as a new paragraph after line 93. This information describes the details of the study location, whereas the introduction should begin with the broad context of the study and then describe the study area. The authors may also want to consider placing some of this information in the methods section as it describes the study location.
Methods for justifying the initial variables selected, their source and how they were calculated from publicly available data are given in good detail.
Results
All figures and graphs are very hard to read, where each pane appears to be squashed. A suggestion is to check an option to retain sizing or resolution when altering figures in Word and should be found as an option in other software.
Discussion
Lines 364-376. All of this is either introduction methods or results. It is reading as a series of statements that need to be placed further up in their relevant sections. This discussion part should explain what caused the variability in model outputs observed, any limiting constraints on the model outputs, and any suggestions to improve model fit in future studies?
Lines 378-380 is also part of the introduction, and the next sentence at lines 380-382 should be placed into results. The discussion needs to state the main results and then explain how this relates to say other modelling done on grasshoppers, other models done on insects or any other habitat in this region.
The next paragraph is just restating results. This now needs to be fixed before the document is further reviewed as it is very difficult to understand the significance of this research with the manuscript in its current form.
Below are some specific comments for consideration
Specific comments
In the summary and the abstract, need to state what country and Provenance or state the Hexi corridor is located in. Is it China?
Lines 15-16 Replace with “MaxEnt model, this study analyzed the main factors affecting the distribution of four grasshopper species that and used this information to predict their suitable habitat area”.
Line 17 Delete “such as namely” and replace with “of”.
Line 20 “Reorder species names as they are normally mentioned in alphabetical order using the genus name to “Angaracris rhodopa, Calliptamus abbreviates, Myrmeleotettix palpalis, and Oedaleus decorus”.
Lines 24-27 Reorder and delete parts of the sentence to change it to “The aim was to provide a basis for grasshopper monitoring, prediction, and precise control. In this study, distribution of suitable habitats for A. rhodopa, C. abbreviates, M. palpalis, O. decorus, with 68, 58, 92 and 61 occurrences recorded by GPS respectively, were predicted under current and future climatic scenarios using a MaxEnt model”
Line 28 “AUC and TSS” please put what they mean in full in the abstract
Line 30. “mean annual precipitation” need to put this in place of “climate” in Line 18.
Line 99. Insert comma after “grasshoppers” and “MaxEnt”
Line 101 Reorder the species names as suggested above and please make this consistent throughout the manuscript. I.e. this applies to the results as well at Lines 240 and after.
Lines 137. Put the graphs as two sets of two panels, as they are difficult to read and cannot see the points at all in the last graphic.
Maybe I have missed some explanation where in methods you show 19 variables though results are only 17? Could you please justify, if not done already, why two of the 19 variables were removed from the analysis.
Line 240. Just noting again the reordering of the species names should be done here.
Line 478 change to “we used 4 grasshopper species in predictions of…”
Need to expand on the significance of the data, for forest management and species conservation.
Comments on the Quality of English LanguageThere are some spelling and grammatical errors throughout this manuscript, though the discussion requires considerable work to relate the results to the broader literature.
Round 2
Reviewer 1 Report
Comments and Suggestions for Authors
Simple Summary and Abstract need to edited for clarity.
For example,
Line 12-13: ...in Gansu, northwest China. Clarifying the distribution range and the main factors influencing their distribution will…
Line 15: Using the MaxEnt model, this study identified the main factors associated with the distribution of four grasshopper species and used this information to predict suitable habitat areas….
Lines 25-27: not clear
Lines 33-34: how much of an increase is this over current known distributions
Lines 42: What are the implications of this modeling?
Introduction.
Line 48: consider rewording…for example,… “Climate-related factors are the primary determinants impacting insect distribution and it is expected that as climate change continues, the range of suitable habitats for insects will changes as well. This shift in suitable habitat ranges has important implications for the impact and management of pest species. For example,…”
Lines 50 -60: The paragraph shift from global climate to abiotic factors on a small scale. Reconsider rewriting this section to more coherently transition from local to global and biotic and abiotic factors
Lines 61-70: Briefly explain what the Hexi Corridor is
Lines 71-76: rewrite for clarity
Line 87: Needs more context / transition
Materials and Methods.
Section needs to be rewritten for clarity
Why are the grasslands important? What are they used for – how important of a pest are grasshoppers to grasslands? This information should be added to the intro
Line 129: What are the surveys? Trapping?
Line 140: A total of 105 what? We sampled a total of 105 locations? How did you sample differently for each species?
Line 210: Not sure 59 is the right citation here
Line 306: Flip the label in Figure 3 to read from Low to High
Line 430-437: I am not following the discussion/logic presented here. Should this section be moved to the methods section? What about the other parameters used?
Lines 443-446: More clearly articulate how this study differs from previous studies, and why
Line 455: What is the threshold for determining when a drought may end?
Line 457-460: Again, I am having a hard time following this logic.
Line 465: NDVI doesn't influence distribution it is a derived remote sensing metric. Include discussion on what NDVI represents in this context. If it was NDVI in a given year, did that correlate with periods of reduced precipitation.
Line 509: Not following this discussion here? How do you get from human intensity to grassland management? Can you please justify?
Line 519-522: Again, not really following the logic here
Line 542: Didn't you argue that precipitation is the primary driver? Why measure these other parameters.
Line 548 Can the authors better justify the value loss of grasshoppers to pastoral areas?
Comments on the Quality of English LanguageThe manuscript needs to be edited for clarity and to better establish justification for conclusions.
Author Response
Thank you again for your comments and hard work on the manuscript! Please refer to the attachment.

Reviewer 2 Report
Comments and Suggestions for Authors
Dear Authors,
all my comments were taken into account. Congratulations on your good work and I wish you good luck.
Best regards
Author Response
Thanks again for your comments and hard work on the manuscript! Good luck with your work!
Reviewer 3 Report
Comments and Suggestions for Authors
Thank you for incorporating the suggestions made to the summary and abstract, as these sections now give a clear understanding of the studies aims and main findings.
Please make a decision to use either “grasshopper” or “locust” throughout the manuscript, and do a global search in the document to ensure consistency throughout the manuscript. The headings are “grasshopper”, as is the title, though “locust” and “grasshopper” are both used in Line 90 and again Lines 243 and 245.
Below are some minor suggestions to further improve the manuscript.
Lines 15-16 Please change to make this sentence clear “….MaxEnt model, this study analyzed whether the main variables of climate, vegetation, soil, topography, and human footprint factors affected the distribution of four grasshopper species and this information was then used to predict their suitable habitat area”.
Line 67 “hm2” the “2” should be superscript
Line 70 “developing methods to reduce the impact of future outbreaks.”
This is indicating that one or more of these grasshopper species reached outbreak levels prior to this study, though this is not mentioned before this remark. If these species did reach outbreak levels this should be explained in the introduction. Also the type of agricultural production needs to be explained somewhere here. It is not entirely clear whether the plant species of the natural grasslands are also grown for agricultural production
Line 254 change to “8.9%, respectively”.
Line 275 please reorder each graph so that the species are shown in alphabetical order ie reorder with figure 2c first followed by 2b,2d then 2a. and do the same order for figure 3 and figure 4 and table 2 and 3 as well.
Lines 425-437 “Provides a suitable habitat for grasshoppers” (need to insert one or several references here)
Lines 546-547“In the context of a future warming climate in which the suitable areas for certain grasshoppers will shrink.”
How, needs expanding, possible change to “will be limited to cooler/warmer areas” or those with “xx conditions”. There needs to be an explanation of the factors contributing this expected change in distribution.
Line 557 It would be good to add a final sentence to show where more research is needed either on these grasshoppers or on potential pests in the Hexi Corridor.
Comments on the Quality of English LanguageIs fine.
Author Response
Thanks again for your comments and hard work on the manuscript!
Please see the attachment.
